# A Fine-Grained Understanding of Uniform Convergence for Halfspaces

**Aryeh Kontorovich** [* 1]   **Kasper Green Larsen** [* 2]

## Abstract

We study the fine-grained uniform convergence behavior of halfspaces beyond worst-case VC bounds. For inhomogeneous halfspaces in $\mathbb{R}^d$ with $d \geq 2$, we show that standard first-order VC bounds are essentially tight: even consistent hypotheses can incur population error $\Theta(d \ln(n/d)/n)$, and in the agnostic setting the deviation scales as $\sqrt{\tau \ln(1/\tau)}$ at true error $\tau$. In contrast, homogeneous halfspaces in $\mathbb{R}^2$ exhibit a markedly different behavior. In the realizable case, every hypothesis consistent with the sample has error $O(1/n)$. In the agnostic case, we prove a bandwise, log-free deviation bound on each dyadic risk band via a critical-wedge localization argument. Unioning over bands incurs only a $\ln \ln n$ overhead, and we establish a matching lower bound showing this overhead is unavoidable. Together, these results give a fine-grained and nearly complete picture of uniform convergence for halfspaces, revealing sharp dimensional and structural thresholds.

## 1. Introduction

Linear models are arguably among the most fundamental learning models and understanding their capabilities and limitations has inspired countless influential theoretical and practical ideas. One of the earliest examples of a learning algorithm is indeed the Perceptron algorithm (Mcculloch & Pitts, 1943) for computing a linear model for binary classification. For binary classification with labels $\{-1, 1\}$, a linear model is specified by a halfspace $h_{w,b}(x) = \text{sign}(w^T x + b)$ where $w \in \mathbb{R}^d$ is a normal vector for the separating hyperplane and $b \in \mathbb{R}$ is the bias.

Understanding the generalization performance of halfspaces in the PAC learning setup of Valiant (1984) is a

core research topic in learning theory. Here there is an unknown target halfspace $h^\star : \mathbb{R}^d \to \{-1, 1\}$ and an unknown data distribution $\mathcal{D}$ over $\mathbb{R}^d$. A training set $S$ is obtained as $n$ i.i.d. samples from $\mathcal{D}$, each labeled by $h^\star$, i.e. $S = (x_1, h^\star(x_1)), \ldots, (x_n, h^\star(x_n))$ with $x_i \sim \mathcal{D}$. The goal is to argue that the error on the training set $S$ for every halfspace $h$, defined as $\text{er}_S(h) = |\{i : h(x_i) \neq h^\star(x_i)\}|/n$, is close to the true error under the distribution $\mathcal{D}$ given by $\text{er}_\mathcal{D}(h) = \mathbb{P}_{x \sim \mathcal{D}}(h(x) \neq h^\star(x))$. If one can give such a guarantee, then this justifies Empirical Risk Minimization where one uses the training data $S$ to find a halfspace $h$ with smallest $\text{er}_S(h)$. Arguing that all halfspaces $h$ have a small gap between $\text{er}_S(h)$ and $\text{er}_\mathcal{D}(h)$ is often referred to as *uniform convergence*, i.e. with enough training data, the performance of every halfspace on the training data $S$ approaches that under the full distribution $\mathcal{D}$.

A classic approach to proving uniform convergence is to use the concept of VC-dimension (Vapnik & Červonenkis, 1971). The VC-dimension of a hypothesis set $\mathcal{H} \subseteq \{-1, 1\}^\mathcal{X}$ for an input domain $\mathcal{X}$, is the largest $d$, such that there exists $d$ points $X = \{x_1, \ldots, x_d\} \subset \mathcal{X}$ where every labeling $y : X \to \{-1, 1\}$ can be realized by a hypothesis $h \in \mathcal{H}$ ($h(x_i) = y(x_i)$ for all $i$). The VC-dimension of halfspaces in $\mathbb{R}^d$ is $d + 1$. This bound allows one to use general uniform convergence results for hypothesis sets of VC-dimension $d + 1$. Concretely, the following result gives a general upper bound on uniform convergence

**Theorem 1.1** (Uniform Convergence for VC-Classes, derived from (Li et al., 2001)). *There is a constant $c > 0$ such that for any input domain $\mathcal{X}$, integer $d \geq 1$, hypothesis set $\mathcal{H}$ of VC-dimension $d$, distribution $\mathcal{D}$ over $\mathcal{X} \times \{-1, 1\}$ any $0 < \delta < 1/2$ and number of samples $n \geq c(d + \ln(1/\delta))$, it holds with probability at least $1 - \delta$ over a sample $S \sim \mathcal{D}^n$ that every hypothesis $h \in \mathcal{H}$ satisfies*

$$|\text{er}_\mathcal{D}(h) - \text{er}_S(h)| \leq$$
$$c \left( \sqrt{\frac{\text{er}_S(h)(d \ln(\frac{e}{\text{er}_S(h)}) + \ln(\frac{1}{\delta}))}{n}} + \frac{d \ln(\frac{n}{d}) + \ln(\frac{1}{\delta})}{n} \right).$$

Note that the result in Theorem 1.1 is more involved than the often quoted and classic $|\text{er}_\mathcal{D}(h) - \text{er}_S(h)| \leq c\sqrt{(d + \ln(1/\delta))/n}$ bound (Blumer et al., 1989). The key difference is that the result in Theorem 1.1 improves for

---

[1] Ben-Gurion University, Beer-Sheva, Israel [2] Aarhus University, Aarhus, Denmark. Correspondence to: Aryeh Kontorovich <karyeh@bgu.ac.il>, Kasper Green Larsen <larsen@cs.au.dk>.

*Proceedings of the 43rd International Conference on Machine Learning*, Seoul, South Korea. PMLR 306, 2026. Copyright 2026 by the author(s).

hypotheses $h$ with small $\mathrm{er}_S(h)$. In the extreme case where $h$ perfectly classifies the training set $S$ ($\mathrm{er}_S(h) = 0$), the bound in Theorem 1.1 simplifies to $c(d\ln(n/d) + \ln(1/\delta))/n$ which is a near-quadratic improvement over the vanilla $c\sqrt{(d + \ln(1/\delta))/n}$ bound. Bounds of the form in Theorem 1.1 are known as *first-order* bounds.

Examining Theorem 1.1, we observe that it applies to *any* hypothesis set $\mathcal{H}$ of VC-dimension $d$. Halfspaces in $\mathbb{R}^{d-1}$ is one example of such a hypothesis set, but it is not a priori clear that the bound in Theorem 1.1 is tight for halfspaces. The terms involving $\ln(1/\delta)$ are known to be tight regardless of the hypothesis set $\mathcal{H}$, but what about the remaining terms?

Quite recently, Hanneke et al. (2024) showed that the bound in Theorem 1.1 is *tight* for *some* hypothesis sets of VC-dimension $d$. Concretely they proved tightness of Theorem 1.1 for the hypothesis set over a finite domain $\mathcal{X}$ consisting of, for every $S \subseteq \mathcal{X}$ with $|S| \leq d$, the hypothesis $h_S$ assigning labels $-1$ to points in $S$ and $+1$ elsewhere. It does not seem possible to choose a finite subset $\mathcal{X}$ of $\mathbb{R}^{d-1}$ such that halfspaces can generate all labelings with up to $d$ points labeled $-1$ (when $|\mathcal{X}|$ is large enough). So we cannot immediately replicate that result. Moreover, the strongest lower bound that holds for *all* hypothesis sets of VC-dimension $d$ states that with constant probability, there is a hypothesis $h$ with $|\mathrm{er}_{\mathcal{D}}(h) - \mathrm{er}_S(h)| \geq c(\sqrt{\mathrm{er}_S(h)d/n} + d/n)$ (Devroye et al., 1996) [Chapter 14]. That is, the $\ln(1/\mathrm{er}_S(h))$ does not appear in the lower bound and neither does the $\ln(n/d)$ term.

In this work, we study precisely this question for halfspaces, i.e. what are the exact uniform convergence guarantees for halfspaces? It turns out that the answer is not so simple and an interesting picture emerges with several surprising theoretical insights.

## 1.1. Our Contributions

Our first main contribution is to show that Theorem 1.1 indeed gives a tight characterization of uniform convergence for halfspaces. However, this comes with a small caveat. In more detail, a halfspace $\mathrm{sign}(w^T x + b)$ is referred to as homogeneous if $b = 0$ and otherwise inhomogeneous. We now have that for inhomogeneous halfspaces in $d \geq 2$ dimensions, we get a tight characterization from Theorem 1.1 as demonstrated by the following two theorems.

**Theorem 1.2.** *There is a constant $c > 0$ such that for any $d \geq 2$ and any $n > d$, there is a distribution $\mathcal{D}$ over $\mathbb{R}^d$ and an inhomogeneous halfspace $h^\star$ such that with probability at least $c$ over a training set $S \sim \mathcal{D}^n$ labeled by $h^\star$, it holds that there is an inhomogeneous halfspace $h$ that is consistent on the training set, i.e. $h(x) = h^\star(x)$ for all $x \in S$, but with $\mathrm{er}_{\mathcal{D}}(h) \geq \frac{cd\ln(n/d)}{n}$.*

**Theorem 1.3.** *There is a constant $c > 0$ such that for any*

$d \geq 2$, *any $c^{-1}d\ln(n/d)/n < \tau < c$ and any $n > c^{-1}d$, there is a distribution $\mathcal{D}$ over $\mathbb{R}^d$ and an inhomogeneous halfspace $h^\star$ such that with probability at least $c$ over a training set $S \sim \mathcal{D}^n$ labeled by $h^\star$, it holds that there is an inhomogeneous halfspace $h$ with $\mathrm{er}_{\mathcal{D}}(h) = \tau, \tau/2 \leq \mathrm{er}_S(h) \leq 2\tau$, but with*

$$\mathrm{er}_{\mathcal{D}}(h) - \mathrm{er}_S(h) \geq c\sqrt{\frac{\mathrm{er}_S(h)d\ln(e/\mathrm{er}_S(h))}{n}}.$$

Observe how Theorem 1.2 matches the upper bound in Theorem 1.1 for $\mathrm{er}_S(h) = 0$, i.e. when $h$ is consistent on the training set. The bound in Theorem 1.3 handles hypotheses with larger error and completely matches the guarantee in Theorem 1.1. Notice also that for $\tau < c^{-1}d\ln(n/d)/n$, the lower bound from Theorem 1.2 matches Theorem 1.1. We remark that the proof of Theorem 1.2 uses a construction very similar to prior work by Zhivotovskiy & Hanneke (2018) (see the proof of their Proposition 19).

A natural question is now whether the restriction to inhomogeneous halfspaces in $d \geq 2$ dimensions is necessary or just an artifact of our proof. It is easily seen that homogeneous halfspaces in $\mathbb{R}^d$ are at least as expressive as inhomogeneous halfspaces in $\mathbb{R}^{d-1}$ using the classic trick of hard-coding the bias into a special feature of value 1 on all data points. But what happens for homogeneous halfspaces in $\mathbb{R}^2$? To our surprise, it turns out that the behavior deviates from the higher-dimensional cases.

**Theorem 1.4.** *For any distribution $\mathcal{D}$ over $\mathbb{R}^2$, any homogeneous halfspace $h^\star$ and any $0 < \delta < 1$, it holds with probability at least $1 - \delta$ over a training set $S \sim \mathcal{D}^n$ labeled by $h^\star$ that every halfspace $h$ that is consistent on the training set, i.e. $h(x) = h^\star(x)$ for all $x \in S$, satisfies $\mathrm{er}_{\mathcal{D}}(h) \leq \frac{\ln(2/\delta)}{n}$.*

Notice how this is an improvement of a $\ln n$ factor over Theorem 1.1 with $d = 2$ (homogeneous halfspaces in $\mathbb{R}^d$ have VC-dimension $d$). Next, for hypotheses with a non-zero $\mathrm{er}_S(h)$, we prove the following generalization upper bound

**Theorem 1.5.** *There is a constant $c > 0$ such that for any distribution $\mathcal{D}$ over $\mathbb{R}^2 \times \{-1, 1\}$, any dyadic risk band $(2^{-i}, 2^{-i+1}]$ with integer $i \geq 1$ and any $0 < \delta < 1/2$, it holds with probability at least $1 - \delta$ over a training set $S \sim \mathcal{D}^n$ that every halfspace $h$ with $\mathrm{er}_{\mathcal{D}}(h) \in (2^{-i}, 2^{-i+1}]$, satisfies*

$$\mathrm{er}_{\mathcal{D}}(h) - \mathrm{er}_S(h) \leq c \cdot \left( \sqrt{\frac{\mathrm{er}_S(h)\ln(1/\delta)}{n}} + \frac{\ln(1/\delta)}{n} \right).$$

Note that in Theorem 1.5, we focus on the general agnostic case where the distribution $\mathcal{D}$ is over a point $x \in \mathbb{R}^2$ *and* a label $y \in \{-1, 1\}$ and $\mathrm{er}_{\mathcal{D}}(h) := \mathbb{P}_{(x,y)\sim\mathcal{D}}(h(x) \neq y)$.

This is an improvement of a $\sqrt{\ln(e/\operatorname{er}_S(h))}$ over the general upper bound provided by Theorem 1.1.

Exploiting that the additive $\ln(1/\delta)/n$ term dominates when $2^{-i} \le 1/n$. A union bound over the $\log_2 n$ relevant dyadic intervals ($i \ge \log_2 n$) allows us to derive the following corollary

**Corollary 1.6.** *There is a constant $c > 0$ such that for any distribution $\mathcal{D}$ over $\mathbb{R}^2 \times \{-1, 1\}$ and any $0 < \delta < 1/2$, it holds with probability at least $1 - \delta$ over a training set $S \sim \mathcal{D}^n$ that every halfspace $h$ satisfies*

$$\operatorname{er}_{\mathcal{D}}(h) - \operatorname{er}_S(h) \le$$
$$c \left( \sqrt{\frac{\operatorname{er}_S(h)(\ln(1/\delta) + \ln\ln n)}{n}} + \frac{\ln(1/\delta) + \ln\ln n}{n} \right).$$

The additive $\ln\ln n$ terms look superfluous at first sight and could easily be suspected of resulting from a sub-optimal union bound. Indeed for Theorem 1.1, the authors prove Theorem 1.1 only for $\operatorname{er}_{\mathcal{D}}(h)$ in a dyadic interval $(2^{-i}, 2^{-i+1}]$ (with the right interpretation of their proof). However, in their case, they can union bound over all dyadic intervals using failure probabilities $\delta_i \approx \delta/(i+1)^2$ since for $\operatorname{er}_{\mathcal{D}}(h) \in (2^{-i}, 2^{-i+1}]$ and $\operatorname{er}_S(h) \approx \operatorname{er}_{\mathcal{D}}(h)$ we see that

$$\sqrt{\frac{2^{-i}(d\ln(2^i) + \ln(1/\delta_i))}{n}} =$$
$$\sqrt{\frac{2^{-i}(d\ln(2^i) + \ln(1/\delta) + 2\ln(i+1))}{n}}.$$

They key point is that $2\ln(i+1)$ is dominated by the $\ln(2^i)$ term and thus can be ignored. This gives the union bound for *free*. However, for our Theorem 1.5 we do not have an additive $\ln(1/\operatorname{er}_S(h))$ term to dominate the additive terms arising from the smaller choice of $\delta$ necessary for a union bound.

In our last contribution, we ask whether the additive $\ln\ln n$ from the union bound is strictly necessary. It turns out it is

**Theorem 1.7.** *There is a constant $c > 0$ such that for $n > c^{-1}$, there is a distribution $\mathcal{D}$ over $\mathbb{R}^2$ and a homogeneous halfspace $h^\star$ such that with probability at least $c$ over a training set $S \sim \mathcal{D}^n$ labeled by $h^\star$, it holds that there is a homogeneous halfspace $h$ with $\operatorname{er}_{\mathcal{D}}(h) \ge c\ln\ln(n)/n$ and*

$$\operatorname{er}_{\mathcal{D}}(h) - \operatorname{er}_S(h) \ge c\sqrt{\frac{\operatorname{er}_S(h)\ln\ln n}{n}}.$$

To the best of our knowledge, this is the first time it has been shown that a simultaneous generalization guarantee over the different risk levels (values of $\operatorname{er}_{\mathcal{D}}(h)$) provably is more expensive than a guarantee over just a single risk level.

## 1.2. Related Work

**PAC learning algorithms, VC theory, and first-order uniform convergence.** While uniform convergence gives a very strong *for all* guarantee, i.e. it bounds $|\operatorname{er}_{\mathcal{D}}(h) - \operatorname{er}_S(h)|$ for every hypothesis $h \in \mathcal{H}$, it is conceivable that concrete learning algorithms may generalize better than what is promised from uniform convergence. If we let $\mathcal{A}(S)$ denote the hypothesis produced by a learning algorithm $\mathcal{A}$ on training set $S$, and we let $h^\star \in \mathcal{H}$ have smallest $\operatorname{er}_{\mathcal{D}}(h)$ among all $h \in \mathcal{D}$, then known lower bounds on the gap $|\operatorname{er}_{\mathcal{D}}(h^\star) - \operatorname{er}_{\mathcal{D}}(\mathcal{A}(S))|$ only scale as $(d + \ln(1/\delta))/n$ and $\sqrt{\operatorname{er}_{\mathcal{D}}(h^\star)(d + \ln(1/\delta))/n}$, i.e. without a $\ln(n/d)$ and a $\sqrt{\ln(1/\operatorname{er}_{\mathcal{D}}(h^\star))}$ factor (Blumer et al., 1989; Ehrenfeucht et al., 1989; Devroye et al., 1996). Work by Simon (1997) and a subsequent improvement by Hanneke (2016) gave the first *optimal* learning algorithm for realizable PAC learning ($\operatorname{er}_{\mathcal{D}}(h^\star) = 0$) whose error scales as $(d + \ln(1/\delta))/n$. This was later matched by several simpler and more natural algorithms (Larsen, 2023; Aden-Ali et al., 2023; 2024; Høgsgaard, 2025). In the agnostic case (general $\operatorname{er}_{\mathcal{D}}(h^\star)$), the recent algorithm by Hanneke et al. (2024) gives the first optimal $\sqrt{\operatorname{er}_{\mathcal{D}}(h^\star)(d + \ln(1/\delta))/n}$ bound provided that $\operatorname{er}_{\mathcal{D}}(h^\star) \ge d\ln^9(n/d)/n$. Subsequent work by Asilis et al. (2025) gave optimal bounds in the $\operatorname{er}_{\mathcal{D}}(h^\star) \le d/n$ regime, leaving only the small range $d/n \le \operatorname{er}_{\mathcal{D}}(h^\star) \le d\ln^9(n/d)/n$ unresolved.

**Support vector machines and margin-based generalization.** Support Vector Machines (SVMs) implement the maximum-margin principle for linear separation, originating with the max-margin classifier Boser et al. (1992) and the soft-margin formulation of Cortes & Vapnik (1995) (see also the monographs Vapnik (1998); Schölkopf & Smola (2002)). A large literature studies generalization in terms of margins, typically via scale-sensitive complexity measures (fat-shattering, covering numbers, Rademacher complexity) and resulting dimension-free or dimension-light bounds; early influential treatments include (Bartlett & Shawe-Taylor, 1999). Recent work has substantially tightened our understanding of *optimal* margin-based guarantees for SVMs, including analyses via geometric Helly-type arguments and stable sample compression, leading to essentially optimal sample complexity bounds for the SVM (Bousquet et al., 2020; Hanneke & Kontorovich, 2021; 2019). In parallel, Grønlund et al. (2020) revisited classic margin bounds for SVMs, proving improved (near-tight) upper bounds together with nearly matching lower bounds that almost settle SVM generalization in terms of margins . Very recently, Larsen & Schalburg (2025) obtained asymptotically tight generalization bounds for large-margin halfspaces, sharpening the classical $\gamma^{-2}$-type dependence in the large-margin regime. These margin-based results are complementary to ours: they provide strong guarantees for *specific* large-

margin predictors (often the maximum-margin solution), whereas our focus is on *uniform* convergence over the full halfspace class without margin restrictions, including consistent hypotheses that may have arbitrarily small margin.

## 2. Homogeneous Halfspaces in the Plane

In this section we prove the upper bounds for homogeneous halfspaces in $\mathbb{R}^2$: Theorem 1.4 (realizable case), Theorem 1.5 (bandwise deviation), and Corollary 1.6 (dyadic union bound). Consider the hypothesis set

$$\mathcal{H} = \{x \to \operatorname{sign}(w^\mathsf{T} x) : w \in \mathbb{R}^2\}$$

consisting of homogeneous halfspaces in $\mathbb{R}^2$.

### 2.1. Realizable Case (proof of Theorem 1.4)

A homogeneous halfspace is $h_u(x) = \operatorname{sign}(u^\top x)$ for some $u \in \mathbb{R}^2$. Since $\operatorname{sign}(u^\top x) = \operatorname{sign}(u^\top (x/\|x\|))$ for $x \neq 0$, the label depends only on the *direction* of $x$. Thus we may push forward $\mathcal{D}$ by $x \mapsto x/\|x\|$ and assume the data lie on the unit circle. Parameterize a point by its angle $\theta \in [0, 2\pi)$. We now think of the data distribution $\mathcal{D}$ as sampling an angle $\theta \in [0, 2\pi)$.

Under this identification, every homogeneous halfspace corresponds to a semicircle. To make boundary effects explicit (and to support distributions with atoms), we fix the convention

$$I_\alpha := (\alpha, \alpha + \pi] \pmod{2\pi},$$

and the classifier outputs $+1$ on $I_\alpha$ and $-1$ on its complement. Assume (w.l.o.g. by rotation) that the target is $I_0 = (0, \pi]$.

For $t \in (0, \pi]$, define the two disagreement sets corresponding to shifts by $+t$ and $-t$:

$$G_t := (0, t] \cup (\pi, \pi + t], \qquad H_t := (\pi - t, \pi] \cup (2\pi - t, 2\pi].$$

Note that $I_t$ disagrees with $I_0$ exactly on $G_t$, and $I_{2\pi - t}$ disagrees with $I_0$ exactly on $H_t$. Also, the families $\{G_t\}_{t \in (0,\pi]}$ and $\{H_t\}_{t \in (0,\pi]}$ are nested: if $0 < s \leq t$ then $G_s \subseteq G_t$ and $H_s \subseteq H_t$.

*Proof of Theorem 1.4.* After pushing forward $\mathcal{D}$ to angles, let $\mu$ denote the induced distribution on $[0, 2\pi)$. Assume the target is $I_0 = (0, \pi]$ as above.

Fix any $\varepsilon \in (0, 1)$. We upper bound the probability that there exists a consistent hypothesis with true error $> \varepsilon$.

*Case 1: shifts by $+t$ (i.e., $\alpha \in [0, \pi]$).* Let

$$t_\varepsilon := \inf\{t \in (0, \pi] : \mu(G_t) \geq \varepsilon\}.$$

By definition and monotonicity, $\mu(G_{t_\varepsilon}) \geq \varepsilon$, and for any $t$ with $\mu(G_t) > \varepsilon$ we have $G_{t_\varepsilon} \subseteq G_t$. Therefore, if there

exists $t$ with $\mu(G_t) > \varepsilon$ and $S \cap G_t = \emptyset$, then necessarily $S \cap G_{t_\varepsilon} = \emptyset$. Thus

$$\begin{aligned}
\mathbb{P}\big(\exists t : \mu(G_t) > \varepsilon \wedge S \cap G_t = \emptyset\big) &\leq \\
\mathbb{P}(S \cap G_{t_\varepsilon} = \emptyset) &= \\
(1 - \mu(G_{t_\varepsilon}))^n &\leq \\
(1 - \varepsilon)^n.
\end{aligned}$$

*Case 2: shifts by $-t$ (i.e., $\alpha \in [\pi, 2\pi)$).* The same argument with the nested family $\{H_t\}$ yields

$$\mathbb{P}\big(\exists t : \mu(H_t) > \varepsilon \wedge S \cap H_t = \emptyset\big) \leq (1 - \varepsilon)^n.$$

By a union bound over the two directions,

$$\mathbb{P}\big(\exists \text{ consistent } h : \mathbb{P}(h \neq h^\star) > \varepsilon\big) \leq 2(1 - \varepsilon)^n \leq 2e^{-n\varepsilon}.$$

Setting $\varepsilon = \frac{1}{n} \ln \frac{2}{\delta}$ gives

$$\sup_{h \in V(S)} \mathbb{P}_{x \sim \mathcal{D}}\big(h(x) \neq h^\star(x)\big) \leq \min\left\{1, \frac{1}{n} \ln \frac{2}{\delta}\right\},$$

where $V(S)$ is the *version space*, i.e., the set of those $h \in \mathcal{H}$ that are consistent with the labeled sample. This implies the stated bound $\ln(2/\delta)/n$. □

*Remark* 2.1. The same tail bound implies the expected worst-case version-space error is $O(1/n)$: if $X = \sup_{h \in V(S)} \mathbb{P}(h \neq h^\star) \in [0, 1]$, then

$$\mathbb{E}[X] \leq \int_0^1 2(1 - \varepsilon)^n \, d\varepsilon = \frac{2}{n + 1}.$$

### 2.2. Agnostic Case

We now generalize the arguments above to the agnostic case. Here $\mathcal{D}$ is a distribution over an angle $\theta \in [0, 2\pi)$ and a label $y \in \{-1, 1\}$ and $\operatorname{er}_\mathcal{D}(h) := \mathbb{P}_{(x,y) \sim \mathcal{D}}(h(x) \neq y)$.

For an integer $i \geq 1$, define the dyadic risk band

$$\mathcal{H}_i := \left\{h \in \mathcal{H} : \operatorname{er}_\mathcal{D}(h) \in (2^{-i}, 2^{-i+1}]\right\}.$$

Assume $\mathcal{H}_i \neq \emptyset$ and fix an arbitrary reference $h' \in \mathcal{H}_i$. By rotation, assume $h' = h_0$ with $I_0 = (0, \pi]$.

For a measurable $A \subset [0, 2\pi)$ and training set $S = (\theta_1, y_1), \ldots, (\theta_n, y_n)$ write

$$\mu(A) := \mathbb{P}_{(x,y) \sim \mathcal{D}}(\theta \in A), \quad \hat{\mu}(A) := \frac{1}{n} \sum_{j=1}^n 1\{\theta_j \in A\}.$$

**Lemma 2.2** (Critical-wedge localization on a band). *Let $h' = h_0$ and let $\varepsilon \in (0, 1)$. Define the critical radii*

$$\begin{aligned}
t_+(\varepsilon) &:= \inf\{t \in (0, \pi] : \mu(G_t) \geq \varepsilon\}, \\
t_-(\varepsilon) &:= \inf\{t \in (0, \pi] : \mu(H_t) \geq \varepsilon\},
\end{aligned}$$

*and the corresponding open wedges*

$$G_\varepsilon^\circ := (0, t_+(\varepsilon)) \cup (\pi, \pi + t_+(\varepsilon)),$$
$$H_\varepsilon^\circ := (\pi - t_-(\varepsilon), \pi) \cup (2\pi - t_-(\varepsilon), 2\pi).$$

*Then $\mu(G_\varepsilon^\circ) \leq \varepsilon$ and $\mu(H_\varepsilon^\circ) \leq \varepsilon$. Moreover, if $h_\alpha$ is a shift with $\alpha \in [0, \pi]$ and $\alpha \geq t_+(\varepsilon)$, then*

$$\mathrm{er}_\mathcal{D}(h_\alpha) \geq \varepsilon - \mathrm{er}_\mathcal{D}(h_0).$$

*The analogous statement holds for shifts in $[\pi, 2\pi)$ using $H_\varepsilon^\circ$.*

*Proof.* If $\alpha \geq t_+(\varepsilon)$ then by nesting $G_{t_+(\varepsilon)} \subseteq G_\alpha$. On $G_{t_+(\varepsilon)}$ the classifiers $h_\alpha$ and $h_0$ always output opposite labels, hence $1\{h_\alpha(\theta) \neq y\} = 1 - 1\{h_0(\theta) \neq y\}$ on that set. Therefore

$$\mathrm{er}_\mathcal{D}(h_\alpha) \geq \mathbb{P}(\theta \in G_{t_+(\varepsilon)}) - \mathbb{P}(h_0(\theta) \neq y) \geq \varepsilon - \mathrm{er}_\mathcal{D}(h_0),$$

using $\mu(G_{t_+(\varepsilon)}) \geq \varepsilon$ by definition. $\square$

**Lemma 2.3** (Bandwise log-free uniform deviation). *Fix $i \geq 1$, assume $\mathcal{H}_i \neq \emptyset$, and fix $h' \in \mathcal{H}_i$. Then for any $\delta \in (0, 1)$, with probability at least $1 - \delta$ over $S \sim \mathcal{D}^n$, simultaneously for all $h \in \mathcal{H}_i$,*

$$\left| \mathrm{er}_S(h) - \mathrm{er}_\mathcal{D}(h) \right| \leq C \left( \sqrt{\frac{2^{-i} \ln(1/\delta)}{n}} + \frac{\ln(1/\delta)}{n} \right),$$

*for a universal constant $C > 0$ (independent of $i, n, \delta$).*

*Proof.* Rotate so $h' = h_0$. We treat the subfamily

$$\mathcal{H}_i^+ := \{h_\alpha \in \mathcal{H}_i : \alpha \in [0, \pi]\},$$

and note the $[\pi, 2\pi)$ case is identical (we union bound over the two cases at the end).

**Step 1: localize all $h \in \mathcal{H}_i^+$ to one wedge.** Set $\varepsilon := 2^{-i+3}$ and let $E := G_\varepsilon^\circ$. Since $\mathrm{er}_\mathcal{D}(h_0) = \mathrm{er}_\mathcal{D}(h') \leq 2^{-i+1}$, Lemma 2.2 implies that any $h_\alpha \in \mathcal{H}_i^+$ must satisfy $\alpha < t_+(\varepsilon)$, hence

$$h_\alpha(\theta) = h_0(\theta) \quad \text{for all } \theta \notin E.$$

In particular, for every $h \in \mathcal{H}_i^+$ we have $h = h_0$ on $E^c$.

Let $p := \mathbb{P}(\theta \in E) = \mu(E) \leq \varepsilon$ and let

$$N_E := |S \cap E| = \sum_{j=1}^n 1\{\theta_j \in E\}.$$

**Step 2: decompose the error inside/outside $E$.** Write conditional (true) errors as

$$\mathrm{er}_\mathcal{D}(h \,|\, E) := \mathbb{P}(h(\theta) \neq y \,|\, \theta \in E),$$
$$\mathrm{er}_\mathcal{D}(h_0 \,|\, E^c) := \mathbb{P}(h_0(\theta) \neq y \,|\, \theta \notin E),$$

and empirical conditional errors analogously on $S \cap E$ and $S \setminus E$. Since $h = h_0$ on $E^c$, we have

$$\mathrm{er}_\mathcal{D}(h) = p\, \mathrm{er}_\mathcal{D}(h \,|\, E) + (1 - p)\, \mathrm{er}_\mathcal{D}(h_0 \,|\, E^c),$$
$$\mathrm{er}_S(h) = \frac{N_E}{n}\, \mathrm{er}_{S \cap E}(h) + \left(1 - \frac{N_E}{n}\right) \mathrm{er}_{S \setminus E}(h_0).$$

A short add-and-subtract gives the deterministic bound

$$\left| \mathrm{er}_\mathcal{D}(h) - \mathrm{er}_S(h) \right| \leq p \left| \mathrm{er}_\mathcal{D}(h \,|\, E) - \mathrm{er}_{S \cap E}(h) \right| + \quad (1)$$
$$\left| \mathrm{er}_\mathcal{D}(h_0 \,|\, E^c) - \mathrm{er}_{S \setminus E}(h_0) \right| + 2\left| p - \frac{N_E}{n} \right|.$$

**Step 3: control each term with probability $1 - \delta$.**

*(a) Control $|p - N_E/n|$.* By Bernstein for Bernoulli indicators, with probability $\geq 1 - \delta/4$,

$$\left| p - \frac{N_E}{n} \right| \leq \sqrt{\frac{2p \ln(4/\delta)}{n}} + \frac{2\ln(4/\delta)}{3n}.$$

*(b) Control the outside-$E$ term for the single hypothesis $h_0$.* Apply Bernstein to the bounded variables $1\{\theta \notin E, h_0(\theta) \neq y\}$ to get, with probability $\geq 1 - \delta/4$,

$$\left| \mathbb{P}(\theta \notin E, h_0(\theta) \neq y) - \widehat{\mathbb{P}}(\theta \notin E, h_0(\theta) \neq y) \right| \leq$$
$$\sqrt{\frac{2\, \mathrm{er}_\mathcal{D}(h_0) \ln(4/\delta)}{n}} + \frac{2\ln(4/\delta)}{3n},$$

where $\widehat{\mathbb{P}}$ denotes the empirical probability measure induced by the sample, and similarly for $|\mathbb{P}(\theta \notin E) - \widehat{\mathbb{P}}(\theta \notin E)|$. Combining these (and using $\mathrm{er}_\mathcal{D}(h_0) \leq 2^{-i+1}$) yields

$$\left| \mathrm{er}_\mathcal{D}(h_0 \,|\, E^c) - \mathrm{er}_{S \setminus E}(h_0) \right| \leq$$
$$c_1 \left( \sqrt{\frac{2^{-i} \ln(1/\delta)}{n}} + \frac{\ln(1/\delta)}{n} \right)$$

for a universal constant $c_1$.

*(c) Control the inside-$E$ uniform term.* Condition on $N_E$ and note that, given $N_E$, the sample in $S \cap E$ is i.i.d. from $\mathcal{D}(\cdot \,|\, \theta \in E)$. The restrictions of semicircles $h \in \mathcal{H}_i^+$ to $E$ form a VC class of constant dimension (in fact $\leq 2$), so a standard VC/Rademacher bound gives, with conditional probability $\geq 1 - \delta/4$,

$$\sup_{h \in \mathcal{H}_i^+} \left| \mathrm{er}_\mathcal{D}(h \,|\, E) - \mathrm{er}_{S \cap E}(h) \right| \leq c_2 \left( \sqrt{\frac{\ln(4/\delta)}{N_E}} + \frac{\ln(4/\delta)}{N_E} \right),$$

for a universal constant $c_2$ (when $N_E = 0$ the left side is 0). Writing $\asymp$ to denote equivalence up to absolute multiplicative constants, we put $p \leq \varepsilon \asymp 2^{-i}$. From the concentration of $N_E$ around $pn$ from (a), we get

$$p \sup_{h \in \mathcal{H}_i^+} \left| \mathrm{er}_\mathcal{D}(h|E) - \mathrm{er}_{S \cap E}(h) \right| \leq c_3 \left( \sqrt{\frac{\ln(1/\delta)}{2^i n}} + \frac{\ln(1/\delta)}{n} \right)$$

with probability at least $1 - \delta/2$, for a universal constant $c_3$.

**Step 4: conclude for $\mathcal{H}_i^+$ and then union bound over the two directions.** Plug the bounds from (a)–(c) into (1) and take a union bound. This yields the stated bound for all $h \in \mathcal{H}_i^+$ with probability at least $1 - \delta/2$. Repeat for the $[\pi, 2\pi)$ case and union bound the two events. $\square$

Note that Lemma 2.3 essentially also gives Theorem 1.7, except we need to argue that we can replace $2^{-i}$ by $\mathrm{er}_S(h)$. Let $c > 0$ be a sufficiently large constant. If $2^{-i} < c \ln(1/\delta)/n$ then we conclude

$$|\mathrm{er}_S(h) - \mathrm{er}_{\mathcal{D}}(h)| \leq$$
$$C\left(\sqrt{\frac{2^{-i} \ln(1/\delta)}{n}} + \frac{\ln(1/\delta)}{n}\right) \leq$$
$$C(\sqrt{c} + 1) \cdot \frac{\ln(1/\delta)}{n} \leq$$
$$C(\sqrt{c} + 1) \cdot \left(\sqrt{\frac{\mathrm{er}_S(h) \ln(1/\delta)}{n}} + \frac{\ln(1/\delta)}{n}\right).$$

If on the other hand $2^{-i} \geq c \ln(1/\delta)/n$, then we have

$$C\left(\sqrt{\frac{2^{-i} \ln(1/\delta)}{n}} + \frac{\ln(1/\delta)}{n}\right) \leq C\left(\frac{2^{-i}}{\sqrt{c}} + \frac{2^{-i}}{c}\right).$$

For large enough constant $c$ compared to $C$, we thus have $|\mathrm{er}_S(h) - \mathrm{er}_{\mathcal{D}}(h)| < 2^{-i-1}$. Since $\mathrm{er}_{\mathcal{D}}(h) \in (2^{-i}, 2^{-i+1}]$, this implies $\mathrm{er}_S(h) \geq \mathrm{er}_{\mathcal{D}}(h) - 2^{-i-1} \geq 2^{-i} - 2^{-i-1} \geq 2^{-i-1}$. Thus we conclude

$$|\mathrm{er}_S(h) - \mathrm{er}_{\mathcal{D}}(h)| \leq$$
$$C\left(\sqrt{\frac{2^{-i} \ln(1/\delta)}{n}} + \frac{\ln(1/\delta)}{n}\right) \leq$$
$$C\left(\sqrt{\frac{2 \, \mathrm{er}_S(h) \ln(1/\delta)}{n}} + \frac{\ln(1/\delta)}{n}\right) \leq$$
$$\sqrt{2} \cdot C\left(\sqrt{\frac{\mathrm{er}_S(h) \ln(1/\delta)}{n}} + \frac{\ln(1/\delta)}{n}\right).$$

This proves Theorem 1.7.

### 2.3. Uniform deviation via a dyadic union bound

*Proof of Corollary 1.6.* Let $m := \lceil \log_2 n \rceil$ and define the tail class

$$\mathcal{H}_\leq := \{h \in \mathcal{H} : \mathrm{er}_{\mathcal{D}}(h) \leq 1/n\}.$$

For $i = 1, \ldots, m$, set $\delta_i := \delta/(i(i+1))$ and set $\delta_0 := \delta/(m+1)$. Then $\sum_{i=1}^m \delta_i = \delta\left(1 - \frac{1}{m+1}\right)$, hence $\sum_{i=1}^m \delta_i + \delta_0 = \delta$.

For each $i \in [m]$, if $\mathcal{H}_i \neq \emptyset$, pick any reference $h_i' \in \mathcal{H}_i$ and apply Theorem 1.7 with confidence $\delta_i$. This gives an event $E_i$ of probability at least $1 - \delta_i$ on which the bound of Theorem 1.7 holds for all $h \in \mathcal{H}_i$. (If $\mathcal{H}_i = \emptyset$, set $E_i$ to be the whole space.)

For the tail class, if $\mathcal{H}_\leq \neq \emptyset$, pick any reference $h_\leq' \in \mathcal{H}_\leq$ and apply the same argument as in Lemma 2.3 with the sole change that we use the uniform upper bound $\mathrm{er}_{\mathcal{D}}(h) \leq 1/n$ for all $h \in \mathcal{H}_\leq$ (the lower endpoint of a dyadic band is not used in the proof). This yields an event $E_\leq$ of probability at least $1 - \delta_0$ on which, simultaneously for all $h \in \mathcal{H}_\leq$,

$$|\mathrm{er}_S(h) - \mathrm{er}_{\mathcal{D}}(h)| \leq c\left(\sqrt{\frac{(1/n) \ln(1/\delta_0)}{n}} + \frac{\ln(1/\delta_0)}{n}\right).$$

(If $\mathcal{H}_\leq = \emptyset$, set $E_\leq$ to be the whole space.)

A union bound yields

$$\mathbb{P}\left(\left(\bigcap_{i=1}^m E_i\right) \cap E_\leq\right) \geq 1 - \sum_{i=1}^m \delta_i - \delta_0 = 1 - \delta.$$

On this intersection, every $h \in \mathcal{H}$ satisfies the desired bound: if $\mathrm{er}_{\mathcal{D}}(h) > 1/n$ then $h \in \mathcal{H}_i$ for some $i \leq m$, and we invoke $E_i$; otherwise $h \in \mathcal{H}_\leq$ and we invoke $E_\leq$. Finally, since $i \leq m = \lceil \log_2 n \rceil$ we have

$$\ln \frac{1}{\delta_i} = \ln\left(\frac{i(i+1)}{\delta}\right) = \ln \frac{1}{\delta} + O(\ln \ln n)$$
$$\ln \frac{1}{\delta_0} = \ln\left(\frac{m+1}{\delta}\right) = \ln \frac{1}{\delta} + O(\ln \ln n).$$

$\square$

## 3. Lower Bounds

Consider inhomogeneous halfspaces in $\mathbb{R}^d$ for even $d \geq 2$. For both our realizable and agnostic lower bound, we design data distributions over a carefully designed finite support $\mathcal{X}_{d/2,k}$. The properties of $\mathcal{X}_{d/2,k}$ are described in the following

**Lemma 3.1.** *For any even $d \geq 2$ and integer $k \geq 1$, there exists $d/2$ sets of points $X_1, \ldots, X_{d/2} \subset \mathbb{R}^d$ so that $|X_i| = k$ for each $i$ and where $\mathcal{X}_{d/2,k} = \cup_{i=1}^{d/2} X_i$ satisfies that any labeling $y : \mathcal{X} \to \{-1, 1\}$ assigning $-1$ to at most one point in each $X_i$ may be realized by an inhomogeneous halfspace in $\mathbb{R}^d$.*

*Proof.* Our construction allocates two coordinates to each $X_i$. For each $i = 1, \ldots, d/2$ let $X_i$ consist of the $k$ points $x_{i,1}, \ldots, x_{i,k}$ where $x_{i,j}$ has all coordinates 0 except coordinate $2i - 1$ that we set to $\cos(j2\pi/k)$ and coordinate $2i$ that we set to $\sin(j2\pi/k)$. We can thus think of $X_i$ as consisting of $k$ points evenly spaced on the unit circle when projected

onto coordinates $2i-1$ and $2i$. Now consider any labeling $y$ of $\mathcal{X}_{d/2,k} = \cup_{i=1}^{d/2} X_i$ assigning $-1$ to at most one point in each $X_i$. We show that $y$ is realized by an inhomogeneous halfspace. We let the *bias* of the halfspace be $\cos(\pi/(4k))$. Note that $0 < \cos(\pi/(4k)) < 1$ for $k \geq 1$. The unnormalized normal vector $w_y$ is chosen so that for every $X_i$ where all points are assigned 1, the two coordinates $2i-1$ and $2i$ are set to 0, and for every $X_i$ where on point $x_{i,j}$ is assigned $-1$, we set coordinate $2i-1$ of $w_y$ to $-\cos(j2\pi/k)$ and coordinate $2i$ to $-\sin(j2\pi/k)$.

Observe that for any $X_i$ where all points are assigned 1 by $y$, we have $\mathrm{sign}(w_y^T x_{i,j} + \cos(\pi/(4k))) = \mathrm{sign}(\cos(\pi/(4k))) = 1$ for all $x_{i,j} \in X_i$. Now for an $X_i$ where one point $x_{i,j}$ is labeled $-1$ by $y$, we have $\mathrm{sign}(w_y^T x_{i,j} + \cos(\pi/(4k))) = \mathrm{sign}(-1 + \cos(\pi/(4k))) = -1$, and for $h \neq j$, we have

$$w_y^T x_{i,h} + \cos(\pi/(4k)) =$$
$$-(\cos(j2\pi/k)\cos(h2\pi/k) +$$
$$\sin(j2\pi/k)\sin(h2\pi/k)) + \cos(\pi/(4k)) =$$
$$-\cos((h-j)2\pi/k) + \cos(\pi/(4k)) \geq$$
$$-\cos(2\pi/k) + \cos(\pi/(4k)).$$

For $k \geq 2$ we have $\cos(2\pi/k) < \cos(\pi/(4k))$ and we conclude $\mathrm{sign}(w_y^T x_{i,h} + \cos(\pi/(4k))) = 1$. $\qquad\square$

Lemma 3.1 will be used to design the support of a data distribution $\mathcal{D}$ in both our realizable and agnostic lower bound. Both lower bounds exploit large deviations in the number of occurrences of a given $x \in \mathcal{X}$ in a sample $S \sim \mathcal{D}^n$ from the expected number of occurrences under. We thus make use of the following two anti-concentration results

**Lemma 3.2** (Klein & Young (2015))**.** *Let $Y_1, \ldots, Y_n$ be independent indicator random variables with success probability $p \leq 1/2$. For every $\sqrt{3/(np)} < \delta < 1/2$,*

$$\mathbb{P}\left(\sum_i Y_i \leq (1-\delta)np\right) \geq \exp(-9np\delta^2).$$

**Lemma 3.3.** *Let $Y_1, \ldots, Y_k$ be negatively correlated indicator random variables, i.e. $\mathbb{E}[Y_i Y_j] \leq \mathbb{E}[Y_i]\mathbb{E}[Y_j]$ for $i \neq j$. Let $Y = \sum_{i=1}^k Y_i$ denote their sum and $\mu = \mathbb{E}[Y]$ its expectation. Then $\mathbb{P}(Y \geq \mu/2) \geq \min\{1,\mu\}/8$.*

*Proof.* We see that

$$\mathbb{E}[Y^2] = \sum_i \sum_j \mathbb{E}[Y_i Y_j]$$
$$\leq \sum_i \mathbb{E}[Y_i^2] + \sum_i \sum_{j \neq i} \mathbb{E}[Y_i]\mathbb{E}[Y_j]$$
$$\leq \mu + \left(\sum_i \mathbb{E}[Y_i]\right)^2 = \mu + \mu^2.$$

By Paley-Zygmund, this implies

$$\mathbb{P}(Y \geq \mu/2) \geq \frac{1}{4} \cdot \frac{\mathbb{E}[Y]^2}{\mathbb{E}[Y^2]} \geq \frac{1}{4} \cdot \frac{\mu^2}{\mu + \mu^2}$$
$$= \frac{1}{4} \cdot \frac{\mu}{1+\mu} \geq \min\{1,\mu\}/8.$$

$\qquad\square$

### 3.1. Realizable Case

We prove our first main lower bound, started in Theorem 1.2.

*Proof of Theorem 1.2.* Let $k := \left\lceil \frac{8n}{d\ln(n/d)} \right\rceil$. Let $\mathcal{D}$ be the uniform distribution over $\mathcal{X}_{d/2,k} = \cup_{i=1}^{d/2} X_i$ and let the target halfspace $h^\star$ be an arbitrary halfspace assigning the label 1 to all points in $\mathcal{X}_{d/2,k}$. Such a halfspace is guaranteed to exist by Lemma 3.1.

We start by arguing that for each $X_i$, it holds with constant probability over a sample $S \sim \mathcal{D}^n$ that there is at least one point $x_{i,j} \in X_i$ with $x_{i,j} \notin S$. For this, define an indicator random variable $Y_{i,j}$ taking the value 1 if $x_{i,j} \notin S$ and 0 otherwise. Then $\mathbb{E}[Y_{i,j}] = \left(1 - \frac{2}{dk}\right)^n \geq \exp\left(-\frac{4n}{dk}\right)$, where the last step uses $\ln(1-p) \geq -2p$ for $p \in [0, 1/2]$ and the fact that $2/(dk) \leq 1/2$ for $k \geq 1$ and $d \geq 2$. By the choice of $k$, we have $dk \geq 8n/\ln(n/d)$ and thus

$$\exp\left(-\frac{4n}{dk}\right) \geq \exp\left(-\frac{\ln(n/d)}{2}\right) = \sqrt{\frac{d}{n}}.$$

Hence $\mathbb{E}[Y_{i,j}] \geq \sqrt{d/n}$ and letting $Y_i = \sum_{j=1}^k Y_{i,j}$ we get $\mathbb{E}[Y_i] \geq k\sqrt{\frac{d}{n}} \geq \frac{8\sqrt{n/d}}{\ln(n/d)}$. In particular, $\mathbb{E}[Y_i] \geq 1$ for all $n > d$. Moreover, the variables $Y_{i,j}$ and $Y_{i,h}$ for $j \neq h$ are negatively correlated, i.e. $\mathbb{E}[Y_{i,j}Y_{i,h}] \leq \mathbb{E}[Y_{i,j}]\mathbb{E}[Y_{i,h}]$. Therefore, by Lemma 3.3,

$$\mathbb{P}(Y_i \geq \mathbb{E}[Y_i]/2) \geq \min\{1, \mathbb{E}[Y_i]\}/8 \geq 1/8,$$

and since $Y_i$ is integer-valued this implies $\mathbb{P}(Y_i \geq 1) \geq 1/8$. Let $Z_i$ be the indicator of the event $\{Y_i \geq 1\}$; then $\mathbb{P}(Z_i = 1) \geq 1/8$. We now have that $\mathbb{E}[\sum_{i=1}^{d/2} Z_i] \geq d/16$. Considering the non-negative random variable $R = d/2 - \sum_{i=1}^{d/2} Z_i$ we have $\mathbb{E}[R] \leq 7d/16$ and thus by Markov's inequality,

$$\mathbb{P}\left[R \leq \frac{15d}{32}\right] \geq 1 - \frac{\mathbb{E}[R]}{15d/32} \geq 1 - \frac{7/16}{15/32} = \frac{1}{15}.$$

On this event we have $\sum_{i=1}^{d/2} Z_i \geq d/32$.

Now define a labeling $y : \mathcal{X}_{d/2,k} \to \{-1, 1\}$ that assigns $-1$ to a point $x_{i,j} \notin S$ for every $i$ where $Z_i = 1$ (choosing one such missing point per such $i$), and assigns $+1$ to all

remaining points. By Lemma 3.1 there is an inhomogeneous halfspace $h_y$ realizing the labeling $y$.

The halfspace $h_y$ labels all points $x_{i,j} \in S$ with the label 1 and is thus consistent with the target $h^\star$ on $S$. However its error under the distribution $\mathcal{D}$ is at least $\mathrm{er}_{\mathcal{D}}(h_y) \geq \frac{d/32}{(d/2)k} = \frac{1}{16k}$. Using the definition of $k$ and the fact that $\lceil A \rceil \leq A + 1 \leq \frac{9}{8}A$ for $A \geq 8$ (and here $A = \frac{8n}{d\ln(n/d)} \geq 8$ since $\ln(n/d) \leq n/d$ for $n > d$), we have $1/k \geq d\ln(n/d)/(9n)$. Therefore,

$$\mathrm{er}_{\mathcal{D}}(h_y) \; \geq \; \frac{1}{16} \cdot \frac{d\ln(n/d)}{9n} \; = \; \frac{d\ln(n/d)}{144n}.$$

This completes the proof for even $d$ by taking $c \leq 1/144$ (and noting the event holds with probability at least $1/15 \geq c$). For odd $d \geq 3$, the lower bound follows from the lower bound for $d' = d - 1$ and a rescaling of $c$ by a factor at most 2. $\qquad\square$

## 3.2. Agnostic Case

We next turn to proving our second lower bound, stated in Theorem 1.3. For this, we need to relate $\mathrm{er}_S(h)$ and $\mathrm{er}_{\mathcal{D}}(h)$ to within a constant factor. We have done this separately in the following lemma

**Lemma 3.4.** *There is a universal constant $c > 0$, such that for any input domain $\mathcal{X}$, integer $d \geq 1$, hypothesis set $\mathcal{H}$ of VC-dimension $d$, distribution $\mathcal{D}$ over $\mathcal{X} \times \{-1, 1\}$, any $0 < \delta < 1/2$ and number of samples $n \geq c(d\ln(n/d) + \ln(1/\delta))$ it holds with probability at least $1 - \delta$ over a sample $S \sim \mathcal{D}^n$ that every hypothesis $h \in \mathcal{H}$ with $\mathrm{er}_{\mathcal{D}}(h) \geq c(\ln(1/\delta) + d\ln(n/d))/n$ has $\frac{1}{2}\mathrm{er}_S(h) \leq \mathrm{er}_{\mathcal{D}}(h) \leq 2\mathrm{er}_S(h)$.*

*Proof.* From Theorem 1.1, it holds with probability at least $1 - \delta$ that every $h \in \mathcal{H}$ satisfies

$$|\mathrm{er}_{\mathcal{D}}(h) - \mathrm{er}_S(h)| \leq$$
$$c'\left(\sqrt{\frac{\mathrm{er}_S(h)(d\ln(\frac{e}{\mathrm{er}_S(h)}) + \ln(\frac{1}{\delta}))}{n}} + \frac{d\ln(\frac{n}{d}) + \ln(\frac{1}{\delta})}{n}\right).$$

for a constant $c' > 0$. Now let $h \in \mathcal{H}$ have $\mathrm{er}_{\mathcal{D}}(h) \geq c(\ln(1/\delta) + d\ln(n/d))/n$ for a sufficiently large constant $c > 0$. We split in two cases. First, if $\mathrm{er}_S(h) \leq \mathrm{er}_{\mathcal{D}}(h)$ then using that $x\ln(e/x)$ is increasing in $x$ for $0 < x < 1$ we see that

$$\sqrt{\frac{\mathrm{er}_S(h)(d\ln(\frac{e}{\mathrm{er}_S(h)}) + \ln(\frac{1}{\delta}))}{n}} + \frac{d\ln(\frac{n}{d}) + \ln(\frac{1}{\delta})}{n} \leq$$
$$\sqrt{\frac{\mathrm{er}_{\mathcal{D}}(h)(d\ln(\frac{e}{\mathrm{er}_{\mathcal{D}}(h)}) + \ln(\frac{1}{\delta}))}{n}} + \frac{\mathrm{er}_{\mathcal{D}}(h)}{c}$$

Using that $\mathrm{er}_{\mathcal{D}}(h) \geq d\ln(n/d)/n$, this is at most

$$\sqrt{\frac{\mathrm{er}_{\mathcal{D}}(h)(d\ln(\frac{en}{d\ln(n/d)}) + \ln(\frac{1}{\delta}))}{n}} + \frac{\mathrm{er}_{\mathcal{D}}(h)}{c} \leq$$
$$\sqrt{\frac{2\mathrm{er}_{\mathcal{D}}(h) \cdot \mathrm{er}_{\mathcal{D}}(h)}{c}} + \frac{\mathrm{er}_{\mathcal{D}}(h)}{c} \leq$$
$$\left(\sqrt{\frac{2}{c}} + \frac{1}{c}\right)\mathrm{er}_{\mathcal{D}}(h).$$

Thus for $c$ large enough, we conclude $|\mathrm{er}_{\mathcal{D}}(h) - \mathrm{er}_S(h)| \leq \mathrm{er}_{\mathcal{D}}(h)/2$, implying $\mathrm{er}_S(h) \geq \mathrm{er}_{\mathcal{D}}(h)/2$. Since we already assumed $\mathrm{er}_S(h) \leq \mathrm{er}_{\mathcal{D}}(h)$ we therefore have $\mathrm{er}_S(h) \leq \mathrm{er}_{\mathcal{D}}(h) \leq 2\mathrm{er}_S(h)$.

Next, if $\mathrm{er}_S(h) > \mathrm{er}_{\mathcal{D}}(h)$, we see that

$$\sqrt{\frac{\mathrm{er}_S(h)(d\ln(\frac{e}{\mathrm{er}_S(h)}) + \ln(\frac{1}{\delta}))}{n}} + \frac{d\ln(\frac{n}{d}) + \ln(\frac{1}{\delta})}{n} \leq$$
$$\sqrt{\frac{\mathrm{er}_S(h)(d\ln(\frac{en}{d\ln(n/d)}) + \ln(\frac{1}{\delta}))}{n}} + \frac{\mathrm{er}_S(h)}{c} \leq$$
$$\sqrt{\frac{2\mathrm{er}_S(h) \cdot \mathrm{er}_S(h)}{c}} + \frac{\mathrm{er}_S(h)}{c} \leq$$
$$\left(\sqrt{\frac{2}{c}} + \frac{1}{c}\right)\mathrm{er}_S(h).$$

For $c > 0$ large enough, we thus have $|\mathrm{er}_{\mathcal{D}}(h) - \mathrm{er}_S(h)| \leq \mathrm{er}_S(h)/2$ hence $\mathrm{er}_S(h) \leq \mathrm{er}_{\mathcal{D}}(h) + \mathrm{er}_S(h)/2 \Rightarrow \mathrm{er}_S(h)/2 \leq \mathrm{er}_{\mathcal{D}}(h)$. We thus have $\mathrm{er}_S(h)/2 \leq \mathrm{er}_{\mathcal{D}}(h) \leq \mathrm{er}_S(h)$. $\qquad\square$

With this established, we are ready to prove Theorem 1.3.

*Proof of Theorem 1.3.* Let $c_1 d\ln(n/d)/n \leq \tau \leq 1/c_1$ for $c_1$ large enough and define $k = 2\lceil d/256 \rceil/(\tau d)$. Assume for simplicity that $k$ is integer (which can be ensured by choosing $\tau$ properly and rescaling $c$ in the lower bound by a constant factor). Let $\mathcal{D}$ be the uniform distribution over $\mathcal{X}_{d/2,k}$ and let the target halfspace $h^\star$ assign the label 1 to all points in $\mathcal{X}_{d/2,k}$.

Proceeding in a similar fashion as the realizable case, we now argue that for a random sample $S \sim \mathcal{D}^n$, a constant fraction of the $X_i$ contains a point $x_{i,j}$ for which $S$ has *few* copies (instead of no copies as in the realizable case). Observe that for any $x_{i,j}$, the number of copies of $x_{i,j}$ in $S$ is binomial distributed with $n$ trials and success probability $2/(dk)$. Now let $Z$ be binomial with $n$ trials and success probability $2/(dk)$. Define $t$ as the largest integer such that $\mathbb{P}(Z \leq 2n/(dk) - t) \geq 1/(8k)$. Using Lemma 3.2 with $\delta = \sqrt{dk\ln(8k)/(18n)}$ shows that $t \geq 2\delta n/(dk) = \sqrt{2n\ln(8k)/(9kd)}$ provided that $\delta$ satisfies the conditions

$\sqrt{3dk/(2n)} < \delta < 1/2$. Since $\delta = \sqrt{dk\ln(8k)/(18n)}$, the first is satisfied for $\ln(8k) > 27$, i.e. when $k$ is a sufficiently large constant. Since $1/(128\tau) \le k \le 2/\tau$, this is ensured by the constraints $\tau \le 1/c_1$ for large enough constant $c_1 > 0$. The second condition $\delta < 1/2$ is satisfied if $\sqrt{d\ln(2/\tau)/(9n\tau)} < 1/2$. This is satisfied by the constraint $c_1 d\ln(n/d)/n \le \tau$ for large enough constant $c_1 > 0$.

Letting $Y_{i,j}$ take the value 1 if we see no more than $2n/(dk) - t$ copies of $x_{i,j}$ in $S$ and $Y_i = \sum_{j=1}^{k} Y_{i,j}$, we have $\mathbb{E}[Y_i] \ge 1/8$. Moreover $Y_{i,j}$ and $Y_{i,h}$ are negatively correlated. Thus by the exact same calculations as in the proof of Theorem 1.2, it holds with constant probability over $S$ that there are at least $\lceil d/256 \rceil$ of the sets $X_i$ (the $\lceil \cdot \rceil$ follows by integrality) that contain at least one point $x_{i,j}$ with no more than $2n/(dk) - t$ copies in $S$.

Now define a labeling $y$ that takes the value $-1$ on an arbitrary set of $\lceil d/256 \rceil$ such points and $+1$ on all remaining points. This labeling is realizable by a halfspace $h_y$ by Lemma 3.1. Examining $h_y$, we first see that $\mathrm{er}_{\mathcal{D}}(h) = \frac{2\lceil d/256 \rceil}{dk} = \tau$. On the other hand, we have

$$\mathrm{er}_S(h) \le \frac{\lceil d/256 \rceil (2n/(dk) - t)}{n} = \mathrm{er}_{\mathcal{D}}(h) - \frac{\lceil d/256 \rceil t}{n}.$$

It follows that

$$\mathrm{er}_{\mathcal{D}}(h) - \mathrm{er}_S(h) \ge \frac{\lceil d/256 \rceil t}{n} \ge c\sqrt{\frac{d\ln k}{kn}}.$$

Since $1/(128\tau) \le k \le 2/\tau$, and $\tau = \mathrm{er}_{\mathcal{D}}(h)$ we have

$$\mathrm{er}_{\mathcal{D}}(h) - \mathrm{er}_S(h) \ge c'\sqrt{\mathrm{er}_{\mathcal{D}}(h)\frac{d\ln(e/\mathrm{er}_{\mathcal{D}}(h))}{n}}.$$

Finally, Lemma 3.4 and a union bound shows that with constant probability, we simultaneously have $(1/2)\mathrm{er}_S(h) \le \mathrm{er}_{\mathcal{D}}(h) \le 2\mathrm{er}_S(h)$ and we may replace $\mathrm{er}_{\mathcal{D}}(h)$ by $\mathrm{er}_S(h)$ in this lower bound. $\qquad\square$

### 3.3. Dyadic Lower Bound for Homogeneous Halfspaces

Consider the point set $\mathcal{X} = \{x_0, \ldots, x_{k-1}\}$ with $x_j = (\cos(2\pi j/k), \sin(2\pi j/k))$ of $k$ points spaced uniformly on the unit circle. We think of $x_j$ as being associated with the angle $2\pi j/k$. The parameter $k$ will be fixed as a power of 2.

Let the target homogeneous halfspace $h^\star$ assign $+1$ to points with an angle $\alpha$ in the semicircle $[0, \pi)$ and $-1$ to the remaining points. Let $\mathcal{D}$ be the uniform distribution over $\mathcal{X}$.

Let $B \le k$ and assume $k/2$ is a power of $B$. For $i = 1, \ldots, \log_B(k/2)$, let $h_i$ be the halfspace corresponding to the semicircle $[2\pi B^i/k, 2\pi B^i/k + \pi)$. The halfspace $h_i$ misclassifies precisely the $2B^i$ points $x_j$ with $j$ in the set $C_i = \{0, \ldots, B^i - 1\} \cup \{k/2, k/2+1, \ldots, k/2+B^i-1\}$.

Define the sets $D_i = C_i \setminus (\cup_{j<i} C_j) = C_i \setminus C_{i-1}$. Then $|D_i| = 2(B^i - B^{i-1})$ (except $|D_1| = 2B$). Since $\mathcal{D}$ is uniform, the expected number of samples a training set $S \sim \mathcal{D}^n$ contains from $D_i$ is $\mu_i = n|D_i|/k$. Now define an indicator random variable $Y_i$ taking the value 1 if $S$ contains fewer than $\mu_i - c\sqrt{\mu_i \ln(\log_B(k))}$ samples from $D_i$ for sufficiently small constant $c > 0$. For $\mu \ge c_1 \ln(\log_B(k))$ for large enough constant $c_1 > 0$, we have $\mathbb{P}(Y_i = 1) \ge 1/\log_B(k/2)$. Furthermore, the $Y_i$ are negatively correlated. Letting $Y = \sum_i Y_i$ it follows from Lemma 3.3 that $\mathbb{P}(Y \ge 1/2) \ge 1/8$. Since $Y$ is integer, this implies $\mathbb{P}(Y \ge 1) \ge 1/8$.

Secondly, a union bound implies that with probability at least $15/16$, we have that $|S \cap D_i| \le \mu_i + c_1\sqrt{\mu_i \ln(\log_B(k))} \le 2\mu_i$ for every $i$, where $c_1 > 0$ is a constant. Now assume $Y \ge 1$ and $|S \cap D_i| \le \mu_i + c_1\sqrt{\mu_i \ln(\log_B(k))} \le 2\mu_i$ for every $i$. These both occur with probability at least $15/16 - 7/8 = 1/16$. Let $i$ be the smallest index such that $Y_i = 1$. Then $|S \cap D_i| \le \mu_i - c\sqrt{\mu_i \ln(\log_B(k))}$. We then have

$$|S \cap C_i| = \sum_{j=1}^{i} |S \cap D_j| \le$$
$$\mu_i - c\sqrt{\mu_i \ln(\log_B(k))} +$$
$$\sum_{j=1}^{i-1}\left(\mu_j + c_1\sqrt{\mu_j \ln(\log_B(k))}\right) \le$$
$$\left(\sum_{j=1}^{i} \mu_j\right) - c\sqrt{\ln(\log_B(k))} \cdot \left(\sqrt{\mu_i} - \frac{c_1}{c}\sum_{j=1}^{i-1} \sqrt{\mu_j}\right).$$

Noting that $\mu_j$ increases by a factory $B$ with $j$, we have for $B \ge 4$ that $\sum_{j=1}^{i-1} \sqrt{\mu_j} \le 2\sqrt{\mu_{i-1}} \le 2\sqrt{\mu_i/B}$. For $B \ge 8(c_1/c)^2$ we conclude $|S \cap C_i| \le \sum_{j=1}^{i} \mu_j - (c/2)\sqrt{\mu_i \ln(\log_B(k))}$. Notice also that $\sum_{j=1}^{i} \mu_j = \mathbb{E}[|S \cap C_i|] = n\mathbb{E}[\mathrm{er}_S(h_i)] = n\mathrm{er}_{\mathcal{D}}(h_i) = 2nB^i/k$. Thus we conclude

$$\mathrm{er}_S(h_i) = |S \cap C_i|/n \le \mathrm{er}_{\mathcal{D}}(h_i) - (c/2) \cdot \frac{\sqrt{\mu_i \ln(\log_B(k))}}{n}.$$

We required $B \ge 8(c_1/c)^2$, so let us fix $B = 8(c_1/c)^2$, which is a constant. For all $i$, we also needed $\mu_i \ge c_1 \ln(\log_B(k))$ for large enough constant $c_1$. Since $\mu_i = n|D_i|/k \ge 2Bn/k$, we can fix $k = c_2 n/\ln\ln n$ for small enough constant $c_2 > 0$. We thus have $\mathrm{er}_{\mathcal{D}}(h_i) - \mathrm{er}_S(h_i) \ge c\frac{\sqrt{\mu_i \ln\ln n}}{n}$ for a constant $c > 0$. Using that $\mu_i \ge |S \cap D_i|/2 \ge |S \cap C_i|/4 = \mathrm{er}_S(h_i)n/4$, this finally gives us Theorem 1.7.

## Acknowledgment

Aryeh Kontorovich is partially supported by the Israel Science and Binational Science Foundations. Kasper Green Larsen is funded by the European Union (ERC, TUCLA, 101125203). Views and opinions expressed are however those of the author(s) only and do not necessarily reflect those of the European Union or the European Research Council. Neither the European Union nor the granting authority can be held responsible for them.

## Impact Statement

This paper presents work whose goal is to advance the field of Machine Learning. There are many potential societal consequences of our work, none which we feel must be specifically highlighted here.

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
