# OpenReview forum: "A Fine-Grained Understanding of Uniform Convergence for Halfspaces"
_ICML.cc/2026/Conference — ICML 2026 regular_

### Official Review · Reviewer_cn9P · 2026-02-22

**Soundness:** 3
**Presentation:** 3
**Significance:** 3
**Originality:** 2
**Overall Recommendation:** 4
**Confidence:** 4

**Summary:**

This paper studies the uniform convergence of halfspaces. In particular, the authors prove upper bounds on the uniform convergence for homogeneous halfspaces in both the realizable and agnostic cases. They also prove lower bounds on the uniform convergence for homogeneous and general halfspaces in both the realizable and agnostic cases.

**Compliance With Llm Reviewing Policy:**

Affirmed.

**Final Justification:**

The authors' response addressed a majority of my concerns, mostly for Thm 1.3. I would like to raise my rating because I believe Thm 1.3 is a sufficiently good finding for halfspaces.

**Key Questions For Authors:**

- Does Theorem 1.3 bring any improvement over Hanneke el al. (2024)?

- Towards the end of the proof of Theorem 1.3, the authors claim $\mathrm{er}_\mathcal{D}(h) = \frac{2\lceil d/256\rceil}{dk}$. However, this seems to only hold in the realizable case where all the labels are generated by the target halfspace $h^*$ (labeled everybody $+1$).  As Theorem 1.3 is concerned with the agnostic case, I do not think this holds. Is there something I missed?

**Limitations:**

I believe the authors could discuss the limitations of their results in applications such as algorithm design.

**Strengths And Weaknesses:**

**Strengths**

- The presentation of this paper is clear and well-organized.

- I verified all the proofs, a majority of they seems sound.

**Weaknesses**

- The major weakness of this work, in my opinion, is the significance of the results:

  - Theorem 1.4, Theorem 1.5, Corollary 1.6, and Theorem 1.7 only apply to $2$-dimensional halfspaces, which is a significantly restricted class;

  - For Theorem 1.3, I do not see any improvement over Theorem 2 in Hanneke et al. (2024), which can yield the same guarantee for halfspaces according to my understanding;

  - Theorem 1.2 and Theorem 1.4 are only applicable to halfspaces under realizable setting, which is also quite limited.

- A minor issue is with the proof writing. The authors seem to have skipped some details. One example is in the proof of Lemma 3.1: it would be much easier to read if the authors explained why it holds that $\cos((h - j)2\pi/k)\leq\cos(2\pi/k)$. Another example is in the proof of Lemma 2.3, where the authors apply Bernstein's inequality twice in part (b) of Step 3. I would advise specifying the parameters instantiated in the inequality and adding Bernstein's inequality in the Appendix for completeness. In the same proof, the authors introduce some unnecessary notation in my view. The aliases $h' = h_0$ and $E:=G_\epsilon^o$ do not help readability, as there are too many symbols for the reader to keep track of. In the proofs of Lemma 3.1 and Theorem 1.2, the authors use $h$ as a subscript, which is already used to denote halfspaces throughout the paper. I suggest avoiding such confusing notation.

- The proof of Theorem 1.3 is potentially flawed, see Questions for details.

[1] Hanneke, S., Larsen, K.G. and Zhivotovskiy, N., 2024, October. Revisiting agnostic PAC learning. In 2024 IEEE 65th Annual Symposium on Foundations of Computer Science (FOCS) (pp. 1968-1982). IEEE.

---

> ### Author Rebuttal · Authors · 2026-03-27
>
> Thank you for your review.
>
> Does Theorem 1.3 bring any improvement over Hanneke et al.?
> Yes. Theorem 2 of Hanneke et al. indeed shows that there EXISTS some hypothesis set where the gap between er_S and er_D must grow as sqrt(er_s ln(1/er_S)). However, Hanneke's work does not say anything about the natural class of half spaces in R^d. See our discussion in line 67 to 88 left. Hanneke's lower bound constructs a particular hypothesis set of VC-dimension d and prove a lower bound for that hypothesis set. However that set does not look like something that corresponds to halfspaces. Thus one cannot infer anything about halfspaces from the work of Hanneke et al. We show that indeed the same growth in error happens for the fundamental class of halfspaces, but ONLY for inhomogeneous in R^2 and up. Theorems 1.4, 1.5 and Cor 1.6 and Theorem 1.7 show that the result of Hanneke et al. actually DOES NOT hold for every hypothesis set of VC dimension d. As such, we believe our collection of results together shows that Hanneke et al.'s work is not the end of the story. Different hypothesis sets of VC-dimension d may behave differently.
>
> Regarding concern with proof of Theorem 1.3:
> If we were proving an upper bound, then indeed there would be a problem if we only gave an upper bound in the realisable case and tried to argue it also holds in the agnostic case. However, since we are proving a lower bound, our lower bound is even stronger when we do not use the additional freedom in the agnostic case to have labels not determined by a hypothesis in H. Since there is even a fixed realisable labelling that causes some half spaces to have a large gap between er_S and er_D, this is "even more" the case if we allow labellings that are not necessarily consistent with a half space. In summary, since we prove a lower bound even without labelling points in a manner not consistent with a half space, the lower bound ALSO HOLDS when points can be labelled in such ways.

---

> > ### Author Rebuttal · Reviewer_cn9P · 2026-04-04
> >
> > I thank the authors for the response. The authors' response addressed a majority of my concerns. Admittedly, the authors' first response does correct my wrong take on Thm 2 of Hanneke et al. (2024). I thought it means there exists a distribution such that their result holds for every VC class of VC dimension $d$. This is my mistake. Indeed, Thm 1.3 does provide different insight than Thm 2 of Hanneke et al. (2024). The second answer addressed my concern about the proof of Thm 1.3. However, for the rest of the theorems, I still think they are too weak because they are on $\mathbb{R}^2$. However, I would like to raise my rating because I believe Thm 1.3 is a sufficiently good finding for halfspaces.

---

### Official Review · Reviewer_dZ3g · 2026-03-10

**Soundness:** 3
**Presentation:** 3
**Significance:** 2
**Originality:** 3
**Overall Recommendation:** 3
**Confidence:** 2

**Summary:**

The paper studies whether the standard first-order VC uniform convergence bound is actually tight for halfspaces. It shows that for inhomogeneous halfspaces in $\mathbb{R}^d$ with $d \ge 2$, the standard first-order picture is essentially correct: there exist distributions for which even a sample-consistent halfspace (zero training loss) has population error on the order of $d \ln(n/d) / n$ (Theorem 1.2).

For the non-zero training loss case, the deviation can scale like $\sqrt{\tau \ln(1/\tau) d / n}$ at risk level $\tau$, although $\tau$ must be somewhat large.

Homogeneous halfspaces in $\mathbb{R}^2$ are studied separately. In the realizable case, the result improves to $O(1/n)$, and in the non-realizable case there is an improvement of order $\sqrt{1 / \mathrm{er}_S(h)}$.

Overall, the contribution is a structural separation between inhomogeneous halfspaces and the special homogeneous planar case, giving a more complete and fine-grained picture of uniform convergence for halfspaces.

**Compliance With Llm Reviewing Policy:**

Affirmed.

**Key Questions For Authors:**

My main conceptual question is about the relation between the lower bounds in Thms 1.2 and 1.3 and the known first-order upper bound in Thm 1.1. The lower bounds appear to be proved under a realizable data-generating model, with labels induced by some $h^\star$, while Theorem 1.1 is an agnostic upper bound over arbitrary distributions. Could the authors clarify in what precise sense these results should be viewed as matching or nearly matching Thm 1.1?

**Limitations:**

yes

**Strengths And Weaknesses:**

The strongest aspect of the paper, in my view, is the set of results for inhomogeneous halfspaces in dimension $d \ge 2$, especially Theorems 1.2 and 1.3. These results show that the standard first-order VC-type bounds are essentially tight even for this important and highly structured hypothesis class, which gives a meaningful fine-grained understanding beyond worst-case VC theory. I think this is the clearest conceptual contribution of the paper: inhomogeneous halfspaces in dimension at least two behave much like worst-case VC classes from the perspective of uniform convergence.

The paper also studies the special case of homogeneous halfspaces in $\mathbb{R}^2$, where it obtains sharper guarantees in both the realizable and agnostic settings. While these results are technically interesting and help complete the overall picture, I found this part somewhat less compelling in terms of broader significance. This may partly reflect my own background, but from my perspective the higher-dimensional inhomogeneous results are the main reason to pay attention to the paper.

On soundness, the results look plausible and mostly well supported. The proofs are relatively compact and clean, which is a positive aspect of the presentation. At the same time, I think the exposition could be improved further. In particular, simple schematic figures illustrating the semicircle representation and the $G_t/H_t$ disagreement regions would make the arguments in the $\mathbb{R}^2$ section substantially easier to follow.

My main reservation is about scope and framing. The paper gives a fairly complete analysis of the planar homogeneous case, but from my perspective it would be more satisfying to have a clearer treatment of what remains open and what has been resolved in the higher-dimensional inhomogeneous setting. In particular, there seem to be some gaps between the upper bound in Theorem 1.1 and the lower bounds in Theorems 1.2 and 1.3. For example, Theorem 1.3 only applies in the regime $\tau \in [c^{-1} d \ln(n/d)/n,, c]$; or another example is that I was not fully convinced by the discussion after Theorem 1.3 claiming that $\operatorname{er}_S(h)$ and $\operatorname{er}_D(h)$ are within constant factors of each other with high probability in the relevant range. For that reason, my main weakness is not with the technical results themselves, but with the paper’s treatment of the remaining gap between the classical upper bounds and the lower bounds proved here.

---

> ### Author Rebuttal · Authors · 2026-03-27
>
> Thanks a lot for your review.
>
> First, regarding the lower bounds in Theorem 1.2 and Theorem 1.3 vs. the upper bound in Theorem 1.1. Let us first remark that the restriction on tau >= dln(n/d)/n is inconsequential. For smaller tau, the tightness follows directly from Theorem 1.2 which gives an example with er_D(h)=0<tau but with |er_D - er_S| at least the bound from Theorem 1.1. We will make this more clear in our writing. Note also that a lower bound in the realisable setting also gives a lower bound in the agnostic setting (it is in that sense a stronger lower bound, whereas an upper in the realisable setting of course wouldn't give an agnostic upper bound).
>
> For the claim on er_D and er_S being within constant factors, we apologies for not giving the full proof. We have added a proof to our manuscript and included the proof in the reply to reviewer hFvA.

---

> > ### Author Rebuttal · Reviewer_dZ3g · 2026-04-04
> >
> > I appreciate the author's rebuttal, but I decide to keep my score.

---

### Official Review · Reviewer_hFvA · 2026-03-12

**Soundness:** 4
**Presentation:** 3
**Significance:** 3
**Originality:** 4
**Overall Recommendation:** 5
**Confidence:** 3

**Summary:**

This work studies generalization guarantees for halfspaces. It demonstrates that known results from VC dimension theory are essentially optimal. The authors examine consistent cases, the agnostic PAC setting, etc. They also analyze the behavior of two-dimensional halfspaces.

**Compliance With Llm Reviewing Policy:**

Affirmed.

**Final Justification:**

The authors have correctly addressed my questions.

**Key Questions For Authors:**

No questions.

**Limitations:**

yes

**Strengths And Weaknesses:**

**Strengths.**

The uniform convergence analysis is thorough. It is made in relation to various known results, and the conclusions are interesting. This work is interested in both proving the optimality of upper bounds and in studying both homogeneous and inhomogeneous halfspaces; this feels complete to me. Completing the study of this class of hypotheses is still relevant today and remains an active topic (through, for example, support vector machines and margin-based generalization).
Although I did not read all five pages of proof thoroughly, the proofs I went through in detail look sound.

**Weaknesses.**

I feel the main weakness is in the theoretical analysis of the obtained results in Section 1.1. For instance, it feels wrong (line 68, right) to treat the distance between $\mathrm{er}\_S(h)$ and $\mathrm{er}\_{\mathcal{D}}(h)$ as being within a constant factor with high probability when $\mathrm{er}\_{\mathcal{D}}(h) \le \tau$ for some $h$; that is to say, a theoretical analysis of any risk can be made using a made-up upper-bounding constant. "When $\mathrm{er}\_{\mathcal{D}}(h)$ is sufficiently larger than $\ln(1/\delta)/n$, we may replace $\sqrt{2-i}$ by $\sqrt{\mathrm{er}_{S}(h)}$" (line 110, left). This is arbitrary as well.

**Typos and small weaknesses.**
1. Line 142: They -> The
2. Line 149/150: Missing end-of-sentence period.
3. Line 215, right: radii

---

> ### Author Rebuttal · Authors · 2026-03-26
>
> Thanks a lot for your review.
>
> The reviewer is (rightfully) concerned about our unproven claims that we may replace $2^{-i}$ by $er_S(h)$ and $er_D(h)$ by $er_S(h)$. We have added proofs for both claims here and have already implemented all these changes in our manuscript and apologies for being sloppy in the submission.
>
> Let us argue that we can replace $2^{-i}$ by $er_S(h)$ in Theorem 1.5.
>
> Let $c>0$ be a sufficiently large constant. If $2^{-i} < c\ln(1/\delta)/n$ then we conclude
> \begin{align*}
> |er_S(h) - er_D(h)| &\leq \\
> C \left(\sqrt{\frac{2^{-i} \ln(1/\delta)}{n}} + \frac{\ln(1/\delta)}{n}\right) &\leq \\
> C(\sqrt{c}+1) \cdot \frac{\ln(1/\delta)}{n}& \leq \\
> C(\sqrt{c}+1) \cdot \left(\sqrt{\frac{er_S(h) \ln(1/\delta)}{n}}+\frac{\ln(1/\delta)}{n} \right).
> \end{align*}
> If on the other hand $2^{-i} \geq c\ln(1/\delta)/n$, then we have
> \begin{align*}
> C \left(\sqrt{\frac{2^{-i} \ln(1/\delta)}{n}} + \frac{\ln(1/\delta)}{n}\right) &\leq C\left(\frac{2^{-i}}{\sqrt{c}} + \frac{2^{-i}}{c}\right).
> \end{align*}
> For large enough constant $c$ compared to $C$, we thus have $|er_S(h)-er_D(h)| < 2^{-i-1}$. Since $er_D(h) \in (2^{-i},2^{-i+1}]$, this implies $er_S(h) \geq er_D(h) - 2^{-i-1} \geq 2^{-i}-2^{-i-1} \geq 2^{-i-1}$. Thus we conclude
> \begin{align*}
> |er_S(h) - er_D(h)| &\leq \\
> C \left(\sqrt{\frac{2^{-i} \ln(1/\delta)}{n}} + \frac{\ln(1/\delta)}{n}\right) &\leq \\
> C \left(\sqrt{\frac{2 er_S(h) \ln(1/\delta)}{n}} + \frac{\ln(1/\delta)}{n}\right) &\leq \\
> \sqrt{2} \cdot C \left(\sqrt{\frac{er_S(h) \ln(1/\delta)}{n}} + \frac{\ln(1/\delta)}{n}\right).
> \end{align*}
>
>
>
> Let us also argue for line 68, right via the following lemma that we have added to the paper:
>
> Lemma
>
> There is a universal constant $c>0$, such that for any input domain $X$, integer $d \geq 1$, hypothesis set $H$ of VC-dimension $d$, distribution $D$ over $X \times \{-1,1\}$, any $0 < \delta < 1/2$ and number of samples $n \geq c(d\ln(n/d) + \ln(1/\delta))$ it holds with probability at least $1-\delta$ over a sample $S \sim D^n$ that every hypothesis $h \in H$ with $er_D(h) \geq c(\ln(1/\delta) + d \ln(n/d))/n$ has
> $
> \frac{1}{2} er_S(h) \leq er_D(h) \leq 2 er_S(h).
> $
>
>
> Proof.
> From Theorem~1.1, it holds with probability at least $1-\delta$ that every $h \in H$ satisfies
> \begin{align*}
> |er_{D}(h)-er_{S}(h)| \leq\\
> c'  \left(\sqrt{\frac{er_{S}(h)(d \ln(\tfrac{e}{er_{S}(h)}) +
> \ln(\tfrac{1}{\delta}))}{n}} + \frac{d \ln(\tfrac{n}{d})+\ln(\tfrac{1}{\delta})}{n}\right).
> \end{align*}
> for a constant $c'>0$. Now let $h \in H$ have $er_D(h) \geq c(\ln(1/\delta) + d \ln(n/d))/n$ for a sufficiently large constant $c>0$. We split in two cases. First, if $er_S(h) \leq er_D(h)$ then using that $x \ln(e/x)$ is increasing in $x$ for $0 < x < 1$ we see that
> \begin{align*}
>     \sqrt{\frac{er_{S}(h)(d \ln(\tfrac{e}{er_{S}(h)}) +
> \ln(\tfrac{1}{\delta}))}{n}} + \frac{d \ln(\tfrac{n}{d})+\ln(\tfrac{1}{\delta})}{n} &\leq \\
> \sqrt{\frac{er_{D}(h)(d \ln(\tfrac{e}{er_{D}(h)}) +
> \ln(\tfrac{1}{\delta}))}{n}} + \frac{er_D(h)}{c} &\leq \\
> \sqrt{\frac{er_{D}(h)(d \ln(\tfrac{e n}{d \ln(n/d)}) +
> \ln(\tfrac{1}{\delta}))}{n}} + \frac{er_D(h)}{c} &\leq \\
> \sqrt{\frac{2er_{D}(h) \cdot er_{D}(h)}{c}} + \frac{er_D(h)}{c} &\leq \\
> \left(\sqrt{\frac{2}{c}} + \frac{1}{c}\right) er_D(h).
> \end{align*}
> Thus for $c$ large enough, we conclude $|er_D(h)-er_S(h)| \leq er_D(h)/2$, implying $er_S(h) \geq er_D(h)/2$. Since we already assumed $er_S(h) \leq er_D(h)$ we therefore have $er_S(h) \leq er_D(h) \leq 2er_S(h)$.
>
> Next, if $er_S(h) > er_D(h)$, we see that
> \begin{align*}
>     \sqrt{\frac{er_{S}(h)(d \ln(\tfrac{e}{er_{S}(h)}) +
> \ln(\tfrac{1}{\delta}))}{n}} + \frac{d \ln(\tfrac{n}{d})+\ln(\tfrac{1}{\delta})}{n} &\leq \\
> \sqrt{\frac{er_{S}(h)(d \ln(\tfrac{en}{d \ln(n/d)}) +
> \ln(\tfrac{1}{\delta}))}{n}} + \frac{er_S(h)}{c} &\leq \\
> \sqrt{\frac{2er_{S}(h) \cdot er_{S}(h)}{c}} + \frac{er_S(h)}{c} &\leq \\
> \left(\sqrt{\frac{2}{c}} + \frac{1}{c}\right) er_S(h).
> \end{align*}
> For $c>0$ large enough, we thus have $|er_D(h)-er_S(h)| \leq er_S(h)/2$ hence $er_S(h) \leq er_D(h) + er_S(h)/2 \Rightarrow er_S(h)/2 \leq er_D(h)$. We thus have $er_S(h)/2 \leq er_D(h) \leq er_S(h)$.

---

> > ### Author Rebuttal · Reviewer_hFvA · 2026-04-01
> >
> > A reformulation of the proofs now involving a constant C is more convincing. I will keep my score.

---

### Official Review · Reviewer_ggcN · 2026-03-13

**Soundness:** 3
**Presentation:** 3
**Significance:** 3
**Originality:** 3
**Overall Recommendation:** 4
**Confidence:** 3

**Summary:**

he paper studies fine-grained uniform convergence for the class of halfspaces, asking how tightly one can control $|\mathrm{er}_D(h)-\mathrm{er}_S(h)|$ as a function of sample size, dimension, and the risk level of $h$, rather than just via the worst-case VC-dimension bound.

The paper’s central question is whether the general bound from Long et. al 2021 is actually the right characterization for geometric halfspaces themselves, rather than for arbitrary VC classes. The paper shows that the answer is nonuniform: for inhomogeneous halfspaces in $\mathbb{R}^d$ with $d\ge 2$, the first-order VC bound is essentially tight, whereas for homogeneous halfspaces in $\mathbb{R}^2$ the behavior is qualitatively better.

For inhomogeneous halfspaces, the paper proves two lower bounds showing that the standard VC-type first-order behavior is unavoidable.

The positive side is the paper’s is more exciting for me: for homogeneous halfspaces in $\mathbb{R}^2$, the authors show that the first-order VC bound is not tight. In the realizable case (Theorem 1.4), every consistent homogeneous halfspace has $er_D \leq \frac {\log (2/\delta)}{n}$ w.p. at least $1- \delta$ improving the bound by $\log n$ factor.

**Compliance With Llm Reviewing Policy:**

Affirmed.

**Final Justification:**

I understood the paper better and I am more confident in my assessment.

**Key Questions For Authors:**

Do you think homogeneous halfspaces in $\mathbb{R}^2$ are the only exceptional case or there could be other natural and more expressive hypothesis class for which Theorem-1.1 is not tight?

The inhomogeneous lower bounds seems highly "discrete" distributions. How robust are they under additional regularity assumptions on $D$? This might be known but I am not aware of such results.

**Limitations:**

Yes

**Strengths And Weaknesses:**

The paper is very well written and quite accessible. Even though I am not a learning theory person and mostly work with TCS and Combinatorial algorithms, I could follow all the high level arguments, results and their impact which I believe is a great positive sign.

Since I do not have an expert knowledge of the area, I am not sure how impressive the results are: the strongest positive result seem somewhat narrow as it is proved only for homogeneous halfspaces in exactly $\mathbb{R}^2$. In addition, how important is it shave log factor for homogeneous halfspaces in $\mathbb{R}^2$? More intellectually stimulating discussion could be nice to add.

Overall, it seems a good result and could be accepted at ICML.

---

> ### Author Rebuttal · Authors · 2026-03-27
>
> We thank the reviewer for the review.
>
> Regarding whether there could be other natural classes, one proposal that comes to our mind would be the fundamental class of decision stumps. These are basically 1d half spaces, but across d different coordinates. As such, we would conjecture that a similar phenomenon would arise there.
>
> Regarding regularity assumptions, we are also not sure. We are not aware of any work that explicitly proposes a regularity assumption that would allow one to avoid our lower bound instance. It could be an interesting direction for future work.

---

> > ### Author Rebuttal · Reviewer_ggcN · 2026-04-03
> >
> > I have more confidence in the paper than I anticipated, I increased my confidence.

---

### Decision · Program_Chairs · 2026-04-30

**Decision:**

Accept (regular)

**Comment:**

The paper studies the problem of learning halfspaces in the (agnostic) PAC model. The popular approach here is to use ERM and analyze it using a uniform convergence argument. However, these uniform convergence arguments typically yield an upper bound that exceeds the (agnostic) PAC lower bound. Motivated by this gap, the authors focus on analyzing the tightness of existing VC-dimension-based uniform convergence bounds when specifically applied to halfspaces.

While it is established that VC-dimension-based uniform convergence cannot yield optimal PAC learners for certain VC classes (as ERM is known to be sub-optimal for them), it was unclear whether halfspaces exhibit this same limitation. The paper resolves this by providing a complete characterization, proving that the sub-optimality of uniform convergence arguments is indeed inherent, even for halfspaces.

The reviewers agree (i) that halfspaces are significant to learning theory (ii) the results are complete and (iii) the paper is very well written.
During the discussion, the reviewers suggested some minor suggestions, which would make the people clearer.
Furthermore, please add discussion how the results in the paper relate to the literature on optimal sample complexity of (agnostic) PAC learning (both in terms of what can be inferred about halfspaces from prior work and what implications this paper has for that literature). Additionally, how does it relate to SVM?

Overall, I recommend the paper be accepted.